# Brain age prediction using multi-hop graph attention module(MGA) with convolutional neural network

**Heejoo Lim**[1,2]                                                                HJHJLIM@EWHAIN.NET

[1] *Division of Mechanical and Biomedical Engineering, Ewha W. University, Seoul, Korea*

[2] *Graduate Program in Smart Factory, Ewha W. University, Seoul, Korea*

**Yoonji Joo**[3]                                                                  YOONJIJOO@EWHA.AC.KR

[3] *Ewha Brain Institute, Ewha W. University, Seoul, Republic of Korea*

**Eunji Ha**[3]                                                                    EUNJIIHA@EWHA.AC.KR

**Yumi Song**[3,4]                                                                YOUME.A.SONG@GMAIL.COM

[4] *Department of Brain and Cognitive Sciences, Ewha W. University, Seoul, Republic of Korea*

**Sujung Yoon**[3,4]                                                              SUJUNGJYOON@EWHA.AC.KR

**In Kyoon Lyoo**[3,4,5,6]                                                        INKYLYOO@EWHA.AC.KR

[5] *Graduate School of Pharmaceutical Sciences, Ewha W. University, Seoul, Republic of Korea*

[6] *The Brain Institute and Department of Psychiatry, University of Utah, Salt Lake City, Utah, United States*

**Taehoon Shin**[1,2]                                                             TAEHOONS@EWHA.AC.KR

**Editors:** Under Review for MIDL 2023

## Abstract

We propose a multi-hop graph attention module (MGA) that addresses the limitation of CNN in capturing non-local connections of features for predicting brain age. MGA converts feature maps to graphs, calculates distance-based scores, and uses Markov property and graph attention to capture direct and indirect connectivity. Combining MGA with sSE-ResNet18, we achieved a mean absolute error (MAE) of 2.822 years and Pearson's correlation coefficient (PCC) of 0.968 using 2,788 T1-weighted MR images of healthy subjects. Our results present a possibility of MGA as a new algorithm for brain age prediction.

**Keywords:** Brain age prediction, graph attention, self attention, deep learning

## 1. Introduction

Deep learning applied to neuroimaging MRI can predict brain age which can serve as a biomarker of brain diseases.(Wang et al., 2019) While CNNs have been applied for brain age prediction, CNN focuses mainly on local features. To overcome this issue, we propose a novel multi-hop graph attention (MGA) module to enhance the performance of CNN. MGA can be flexibly applied to any type of CNN architecture and exploits direct and indirect connections among non-local feature domain regions in the middle of convolution layers. MGA was combined with sSE-ResNet18(Roy et al., 2018) for the final model, which achieved the lowest MAE compared to other computer vision algorithms.

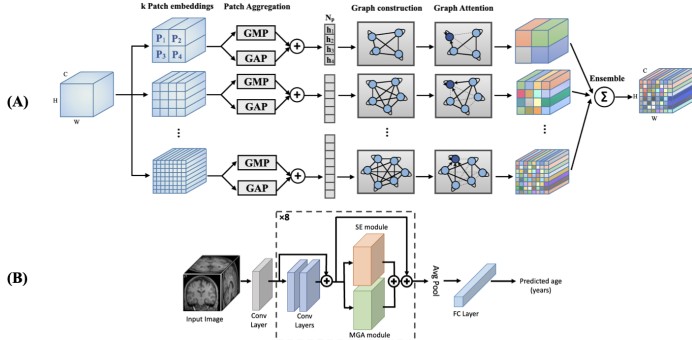

Figure 1: (A) An overview of the proposed multi-hop graph attention (MGA) module. (B) Model structure of MGA-sSE-ResNet18 for brain age prediction.

## 2. Method

A schematic diagram of the proposed MGA module is shown in Fig1(A). Placed in between convolution layers, the MGA module first constructs a graph structure by defining nodes and edges. After parcellating the feature map with a $N_p$ patches using the patch ratio $\gamma$, which is a hyperparameter, we aggregate all spatial-channel dimensions of each patch by using global average pooling (GAP) and global max pooling (GMP). The two pooled tensors are then concatenated and serve as a node set: $\mathbf{H}=\{\mathbf{h}_1, \mathbf{h}_2, ..., \mathbf{h}_{N_p}\}$, $\mathbf{h}_i \in \mathbb{R}^2$. We then define the edge $e_{ij}$ between the two nodes $\mathbf{h}_i$ and $\mathbf{h}_j$ as follows.

$$e_{ij} := 1/(exp(\|\mathbf{V}\mathbf{h}_i - \mathbf{V}\mathbf{h}_j\|_2)) \tag{1}$$

The use of the learnable embedding $\mathbf{V}(\in \mathbb{R}^{2\times 2})$ is because the connections of patches are more complex than computing the direct similarity of image features. We obtain $m$-hop edge matrix $\mathbf{E}_\forall^{(m)}$ using the Markov property as follows.

$$\tilde{\mathbf{E}}_\forall^{(m)} = \tilde{\mathbf{E}} + \beta\tilde{\mathbf{E}}^2 + \beta^2\tilde{\mathbf{E}}^3 + ... + \beta^{m-1}\tilde{\mathbf{E}}^m = \sum_{k=1}^{m} \beta^{k-1}\tilde{\mathbf{E}}^k, \quad \mathbf{E}_\forall^{(m)} := (\tilde{\mathbf{E}}_\forall^{(m)} + \tilde{\mathbf{E}}_\forall^{(m)T})/2 \tag{2}$$

Note that $m$ and $\beta$ are also hyperparameters that determine hop size and multi-hop weight, respectively. The formed graph sets go through a graph attention block with a gate operation that updates the patch set based on self-attention and obtains an updated feature map. This procedure can be repeated in parallel for $k$ sets of patches, and the resulting feature updates are integrated to produce the final output of the module. We combined MGA with sSE-ResNet18 for our final prediction model, as shown in Fig1(B).

## 3. Experiment and Result

Three-dimensional T1-weighted MR images of 2,788 healthy subjects(age:20-70years) were obtained from 7 public datasets: OpenNeuro, COBRE, Open fMRI, INDI, IXI, FCP1000, and XNAT. We randomly divided samples into three subsets: 1) the training dataset (1951

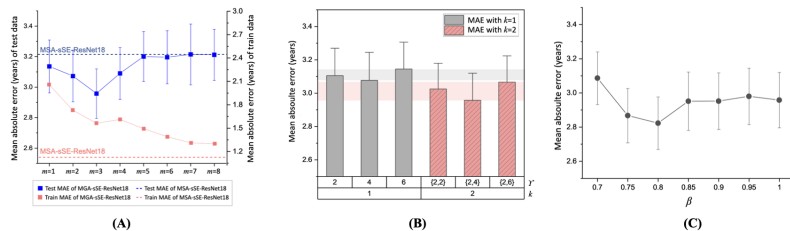

**(A)**                               **(B)**                               **(C)**

Figure 2: (A): Effect of hop size $m$.(blue:test error, red:train error) (B): Effect of $\gamma$ and $k$.(gray:$k$=1, red:$k$=2) (C): Effect of multi-hop weight $\beta$.

| Model | MAE | PCC |
|---|---|---|
| ResNet18 | 3.249 | 0.948 |
| sSE-ResNet18 | 3.239 | 0.956 |
| DenseNet121 | 3.340 | 0.961 |
| SFCN | 3.233 | 0.949 |
| TSAN | 2.892 | 0.956 |
| MSA-sSE-ResNet18 | 3.216 | 0.960 |
| MGA-sSE-ResNet18 | 2.822 | 0.968 |
| MGA-ResNet18 | 3.065 | 0.955 |

(1) Performance comparison with other SOTA models.

(2) Scatter plots of (A) MGA-sSE-ResNet18, and (B) sSE-ResNet18.

Figure 3: Comparison of model performance and scatter plots of the proposed model.

samples), 2) the validation dataset (419 samples), and 3) the test dataset (418 samples). We first examined the key parameters of MGA, which is hop size $m$, patch ratio $\gamma$, number of branches $k$, and multi-hop weight $\beta$ where the results are displayed in Fig2. Fig2(A) shows that test MAEs of MGA with $m$<5 are lower than MAE of multi-head self-attention (MSA), indicating that it is beneficial to consider important embeddings only rather than all when calculating the self-attention coefficients. The final network was chosen based on the performance of the validation dataset. We also compared our model with 5 different CNN models, where SFCN(Peng et al., 2021) and TSAN(Cheng et al., 2021) is the state-of-the-art model in brain age prediction field. In Fig3(1), MGA-sSE-ResNet18 achieved the lowest MAE(2.822 years) and highest PCC(0.968) among the comparisons. Other prediction models, such as Vision Transformer(ViT) or Graph attention network(GATs) were also evaluated, but showed a poor performance, presumably due to insufficient training data. It is also shown that implementing the MGA module reduces model bias and variance (Fig3(2)). From the results, we have shown that interleaving MGA with conventional CNN can improve accuracy and thus effective for brain age prediction.

## Acknowledgment

This research was supported by Basic Science Research Program through the National Research Foundation of Korea (NRF) funded by the Ministry of Education (NRF-2020R1A6A1A03043528), and Institute of Informationcommunications Technology Planning Evaluation (IITP) grant (No. RS-2022-00155966, Artificial Intelligence Convergence Innovation Human Resources Development).

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
