# OpenReview forum: "Brain age prediction using multi-hop graph attention module(MGA) with convolutional neural network"
_MIDL.io/2023/Short_Paper_Track — MIDL 2023 Short paper track Poster_

### Official Review · Reviewer_8dS4 · 2023-04-19
**Well motivated, validation might have been stronger**

**Rating:** 7
**Confidence:** 5

**Review:**

This paper propose the introduction of a graph attention module, which can be introduced between convolutional layers of any convolutional network. To achieve this the model parcellates up the convolutional feature maps and then learns a graph, by assigning edges according to distances in an embedding space. This is passed to a graph attention network to learn long range associations between features, and in this way the approach is shown to improve prediction on a brain age prediction task.

Strengths:
Clearly written
Well motivated
Shows improvement over previously published methods

Weaknesses:
One has to wonder why they didn’t just try an attention u-net or a vision transformer
There are probably better methods to benchmark against, for example Bass C TMI 2022, or Dinsdale Neuroimage 2021

Overall I think its an interesting idea and a solid enough paper to be accepted

---

### Official Review · Reviewer_w16w · 2023-04-22
**Interesting approach with thorough experimental settings tested**

**Rating:** 7
**Confidence:** 4

**Review:**

This paper proposes a multi-hop graph attention (MGA) model which was combined with a CNN for brain age prediction from T1-weighted MRI. The approach was tested on a single split of a large dataset of almost 3K images from 7 public datasets.

Strengths:
+ Multi-hop attention is an interesting approach to incorporate longer range relationships and is a standalone module that could be applied to many networks
+ Extensive experimental settings were tested - variation in important hyperparameters, comparison to more standard self-attention, ablation of MGA, comparison to many other brain prediction methods.
+ Results suggest potential improvement compared to standard self-attention and state-of-the-art approaches

Weaknesses:
- Unclear how the proposed MGA module differs from other multi-hop attention graph methods (ie. Wang et al., Multi-hop Attention Graph Neural Networks, IJCAI 2021) - should include a statement /discussion
- Some unclear methods details - e.g., how exactly is validation data used
- Statistical analysis of results would be helpful to understand the significance of differences between methods because the nominal size of the differences i small